# Social Media Usage by Different Generations as a Tool for Sustainable Tourism Marketing in Society 5.0 Idea

Beata Hysa [1,*], Aneta Karasek [2] and Iwona Zdonek [1]

[1] Department of Organization and Management, Institute of Economy and Informatics,
   Silesian University of Technology, Akademicka 2A, 44-100 Gliwice, Poland; iwona.zdonek@polsl.pl
[2] Faculty of Economics, Maria Curie-Skłodowska University, Plac Marii Curie-Skłodowskiej 5,
   20-031 Lublin, Poland; aneta.karasek@umcs.pl
*  Correspondence: beata.hysa@polsl.pl

**Abstract:** This article discusses the use of social media by different generations in destination marketing from a sustainable tourism perspective. In the light of the global COVID-19 pandemic, intensive marketing efforts and strategies to bring back sustainable tourism will soon become important. Social media (SM) can significantly support the promotion of destinations by guaranteeing an appropriate number and type of tourists. The article examines the frequency of using social media by different generations and the scope of their use in planning a tourist trip. The research was conducted in Poland on a sample of 397 respondents representing the group of Baby Boomers (BB), as well as Generations X, Y, and Z. The results of the research showed that the frequency of using SM decreases with age. The differences between the generations are visible in such behaviours as using SM to check opinions about tourist places, recommending a holiday with positive opinions and comments in SM, as well as resigning from a holiday based on negative opinions and comments.

**Keywords:** social media; sustainable tourism marketing; restart tourism; different generations; Society 5.0 idea

## 1. Introduction

Due to the great importance and impact of tourism on society and the environment, governments, public institutions, and commercial companies more and more often see the need to emphasise the positive effects of tourism and limit its negative consequences. [1]. This intention is also in line with the idea of Society 5.0. The concept of Society 5.0 aims to create a smart society, in which the integration of cyberspace and the real world using state-of-the-art technologies helps different sectors, countries, and regions to cooperate with each other in such a way as to achieve the goals of sustainable development [2]. It is, therefore, necessary to achieve a balance between sustainable development and the development of tourism or to restore this tourism after the Covid-19 pandemic [3]. Tourism activity is inextricably linked with the surrounding environment; therefore, it is necessary to raise awareness among tourists, residents, and managers of tourist facilities about the sustainable development of the tourism industry by adapting sustainable practices. This also applies to marketing activities. For the tourism industry, an appropriate marketing strategy is responsible for the number and type of tourists visiting a given place and for guaranteeing profitable destinations that make a valuable contribution to the economic development of a given country [4].

At the beginning of the twenty-first century, most enterprises and organisations used the Internet for one-way communication, which is comparable to traditional promotion tools, such as advertising [5,6]. Nowadays, thanks to social media, the use of the Internet in marketing communication is changing significantly. User-generated content on various social networks largely determines the image of an organisation, even influencing its revenues and, as a result, survival. New opportunities for communication with users are also

available to organisations in the tourism industry, where they are used in the marketing of a new tourism product. Social media are effective information and communication channel for various generations of tourists [7–9]; therefore, they should be used by marketers (hotels, restaurants, city authorities) to effectively reach potential recipients with the promotion of new places. Creating a good, effective promotional offer on social media should encourage users to visit a given place. Moreover, the use of social media becomes extremely important in the course of a sustainable return to tourism after the SARS-CoV-2 virus pandemic, which requires the restoring of trust through communication [10].

Although you can more and more often find in the literature publications on the use of social media by various groups of generations [11–14], there is little research focusing on the use of this medium in the promotion of places and the return of tourism, to maintain its sustainability. However, with the help of social media, tourists of different generations more and more often plan their trips and tourist journeys [9]. Therefore, in the presented article, the main goal is to identify the activity, behaviour, and preferences of various generations in social media when travel (holidays) planning. Pursuing this goal will allow the development of new marketing strategies for the use of social media in the sustainable recovery of tourism.

The article was organised as follows. Section 2 presents the current knowledge in the field of sustainable development, sustainable marketing, the idea of Society 5.0, and the use of social media in tourism from the perspective of various generations of tourists. Section 3 describes the methodology of empirical research. The research results and discussion are presented in Section 4, whereas conclusions, limitations, and further research is covered in Section 5.

## 2. Literature Review

### 2.1. Sustainable Development and Sustainable Tourism Marketing

In recent years, the dynamic development of the tourism industry has been observed, which was influenced by the emergence of new forms of travel, the expansion of the list of tourist destinations, and an increase in the world's population. This industry was an important sector of the economy, as it is estimated at 7% of global trade [3], and tourists' expenses in 2019 amounted to USD 1397.6 trillion [3]. Additionally, an increase in tourist traffic and the frequency of holidays were observed. In 2019, 1460 million international tourist arrivals were carried out, 51% of which concerned European countries. Among the growing population, the needs related to active leisure time, relaxation, and discovering new destinations were growing, which had a significant impact on both the tourism sector and the local population. In turn, in 2020, the situation related to the COVID 19 pandemic caused turbulence on the tourist market and a dramatic reduction in the number of travels. In 2020, there was a drop between 850 million to 1.1 billion international tourists, while the largest was recorded in March and April of 2020, as it was −97% compared to the previous year [3].

The tourism sector is currently experiencing a transformation caused by Covid-19, technology and digital economy, customer demographics, customer values and habits, as well as an economic imperative [3]. Currently, there are recommended actions at many levels to restore tourism. The United Nations World Tourism Organization (UNWTO) points to the need for the sustainable recovery of tourism, the elements of which are to include exploring the emotional and economic capacity of customers to travel, as well as accelerating the digital transformation [3]. These factors make it necessary to redefine the methods of tourism and to meet the increased expectations of travellers. To meet the contemporary requirements, it is crucial to remember the compliance with and implementation of the Sustainable Development Goals (SDGs).

In recent years, we have seen an increase in interest in the implementation of the sustainable development goals and actions taken by governments, nongovernmental organisations, and enterprises in order to achieve them. Sustainability principles refer to the environmental, economic, and socio-cultural aspects of tourism development—and a

suitable balance must be established between these three dimensions to guarantee its long-term sustainability [15]. While tourism may have a positive impact on a given destination, it may also have a negative impact on the environment and the local community [1]. Therefore, it is necessary to take measures aimed at environmental and community protection, which should be implemented by various stakeholders. However, difficulties are observed for tourism enterprises to adopt a business model based on sustainable paradigms, such as the circular economy [16]. Support in implementing SDG goals should be provided by prudent public policy, where host governments must strive to promote socially and environmentally responsible tourism industries in their respective countries [1]. On the other hand, the customers themselves are becoming more and more aware of the need to take care of the environment. Fifty-eight percent of consumers say they are thinking more about the environment since COVID-19 [17] (World Travel and Tourism Council, 2020). This indicates that public awareness in this area is increasing and that it increases the cost of maintaining the facilities. Research indicates that respondents are ready to pay under the influence of individual pro-environmental attitudes and beliefs, as well as make donations if tourist facilities are characterised by the highest standard of sustainable development [18].

The implementation of the SDG goals in the field of tourism is undertaken at the governmental and nongovernmental level, as well as by the entrepreneurs themselves. These activities are aimed at the sustainable use of available natural resources and the implementation of sustainable management practices in the enterprise. In order to implement activities in the area of sustainable tourism, it is essential to ensure socio-economic benefits to all stakeholders. This includes stable employment and earning opportunities, as well as social services for host communities and contributing to poverty reduction [15]. This is a big challenge for the tourism industry, to ensure—on the one hand—the greatest possible tourist offer for customers, bearing in mind their diverse expectations and needs, but also to encourage them to change their behaviour in favour of sustainable development through conveying messages to them [15].

Identifying more persuasive methods of communication in order to achieve a change in behaviour in tourists regarding their involvement in sustainability is used more and more in tourism marketing [19]. It is indicated in the literature that actions in the area of social influence, habit formation, the individual self, feelings, and cognition, as well as tangibility may contribute to a permanent change in consumer behaviour [20]. This indicates that actions should be taken in the above-mentioned areas, and it is also worth ensuring that they are consistent. Actions taken under sustainable tourism marketing should also be directed to tourists, for whom sustainable development is important. Then, advertisements targeting this consumer segment tend to emphasise the biospheric-altruistic aspects [21]. Research indicates that consumers, who are highly interested in sustainability, are persuaded by messages that include details about the hotel's sustainability performance, to increase the social-environmental well-being, whereas, for customers, who are less interested in sustainability, a self-referential emotional communication is essential, as it increases the emotional well-being [21]. Therefore, when undertaking marketing activities and bearing in mind the diversified approach of tourists to sustainable development, their approach to sustainable development should be taken into account in order to create an appropriate communication message.

As indicated by UNWTO, it is now necessary to undertake activities aimed at restarting tourism, for which a targeted marketing and promotion campaign should be used [22]. Destinations shall send clear and consolidated messages to their source markets and adjust to their perceptions and needs to regain visitor confidence, given the importance and current sensitivities towards public health [10]. Therefore, tourism slogans and logos distributed among six clusters—spiritual serenity, symbolic image, emotional flow, natural discovery, creative aesthetics, and cultural experience—should be included in the marketing strategy [23].

Currently, tourists—while planning holidays—use new technologies that help them obtain information about a given place, services offered, and tourist attractions. Conducting marketing activities is aimed at encouraging potential tourists to take advantage of the company or region's offer. Along with technological development, organisations more and more often use new technologies in this area, including social media. In 2019, the costs of digital advertising in the travel industry reached USD 5.5 trillion, representing 4.2% of total digital spending [24]. This indicates a high value of expenses on conducting marketing activities, in which expenses on activities in the digital area are important. In the travel industry, 45% of advertisement costs went towards digital marketing in 2019, while 47% of travel marketers maintain a continuous digital presence [25]. It is forecasted that the tourism sector will allocate more and more funds towards expenses in search networks than other sectors. Search networks will account for 58.8% of its digital budget, compared with the industry-wide figure of 40.4% [26]. In order to effectively reach tourists with the offer, it is necessary to bear in mind the need to use tools that will help potential tourists understand the tourist resources of a given region. Among them, social media are used [27].

It seems necessary to undertake marketing activities to encourage customers to visit tourist attractions, however, under new rules and taking into consideration their preferences. In order to undertake integrated actions in this area, it is necessary to get acquainted with the conditions of Society 5.0 and the tools that are used by it.

### 2.2. Society 5.0 Idea

The greatest challenge in pursuing the concept of sustainable development today is to create a comprehensive system, in which all nations work together for a sustainable world, that can both achieve economic development, as well as invent solutions to social problems [28]. The idea of Society 5.0 is to help in the implementation of 17 goals (Sustainable Development Goals, SDGs) set out in the 2030 Agenda for Sustainable Development by UN countries.

Society 5.0 is a Japanese proposal for the concept of a modern, future-oriented, and human-centric society, in which the integration of cyberspace and the real world is to be implemented using the latest technologies, such as artificial intelligence, Internet of Things, robotics, or big data. In 2016, the Japanese government presented the vision of a "super-smart society" in its 5th Science and Technology Basic Plan dedicated to innovation and digitisation [29]. In turn, in 2017, the Advisory Council for the Promotion of Science and Technology Diplomacy, chaired by the Minister of Science and Technology at the Minister for Foreign Affairs of Japan, developed recommendations for the science and research sector, as it is this sector that bears the greatest burden of providing solutions to global problems. The recommendations set out the contribution that Japan should make to the achievement of the Sustainable Development Goals (SDGs) through Science, Technology, and Innovation (STI) [28]. This document identifies four key initiatives for the Japanese science and research sector [30]:

- creating a global future through Society 5.0,
- enabling solutions using global data,
- promoting cooperation at global and cross-sectoral level,
- supporting human resources to undertake science, technology, and innovation (STI) efforts in support of the Sustainable Development Goals.

Thus, it can be concluded that science, research, and innovation are treated as intermediary forces connecting various sectors, countries, and regions, the cooperation of which should contribute to the achievement of the Sustainable Development Goals. It is worth noting that similar assumptions underpin the European concept of Responsible Research and Innovation (RRI), which assumes that research and innovation should take into account complex reality and respond to contemporary challenges [28]. It is proposed to distinguish internal and external stakeholders, which makes it possible to identify factors that play an important role in the implementation of actions in the area of responsible innovation.

Science and technology-based innovation has already triggered some changes in society, but these changes can only be positive if society is ready for them. Today's digital society is much more complex and interconnected than ever. For Society 5.0 to be created for the purposes of sustainable development, one must consciously choose to collaborate with stakeholders from a variety of knowledge domains, including economics, social sciences, and the humanities [30].

Society 5.0 aims to create a world, where essential goods and services are provided to anyone, at any time and place, regardless of region, age, gender, language, or other restrictions. It aims to achieve economic growth and well-being at the same time, as well as overcome social challenges, and, thus, contribute to the well-being of the global community [28]. Such a society is also called a super-intelligent society or a creative society, and is another society, after the hunting society, agrarian, industrial, and the current information society [31]. Society 5.0 suggests using advanced technologies and products to connect people and things, as well as share all kinds of knowledge and information in creating new social and business chains and values in society [32]. At the moment, employees work in conditions characterised by information overflow, and, thus, the search and analysis of information is difficult and is not adequately supported by available technological solutions. Compared to information Society 4.0, Society 5.0 is made up of creative, intelligent people, who—with the help of digital technologies and data—lead a diverse lifestyle and strive for happiness in their own way. Proposed changes that should be made in Society 4.0, so that it could be called Society 5.0, are shown in Figure 1.

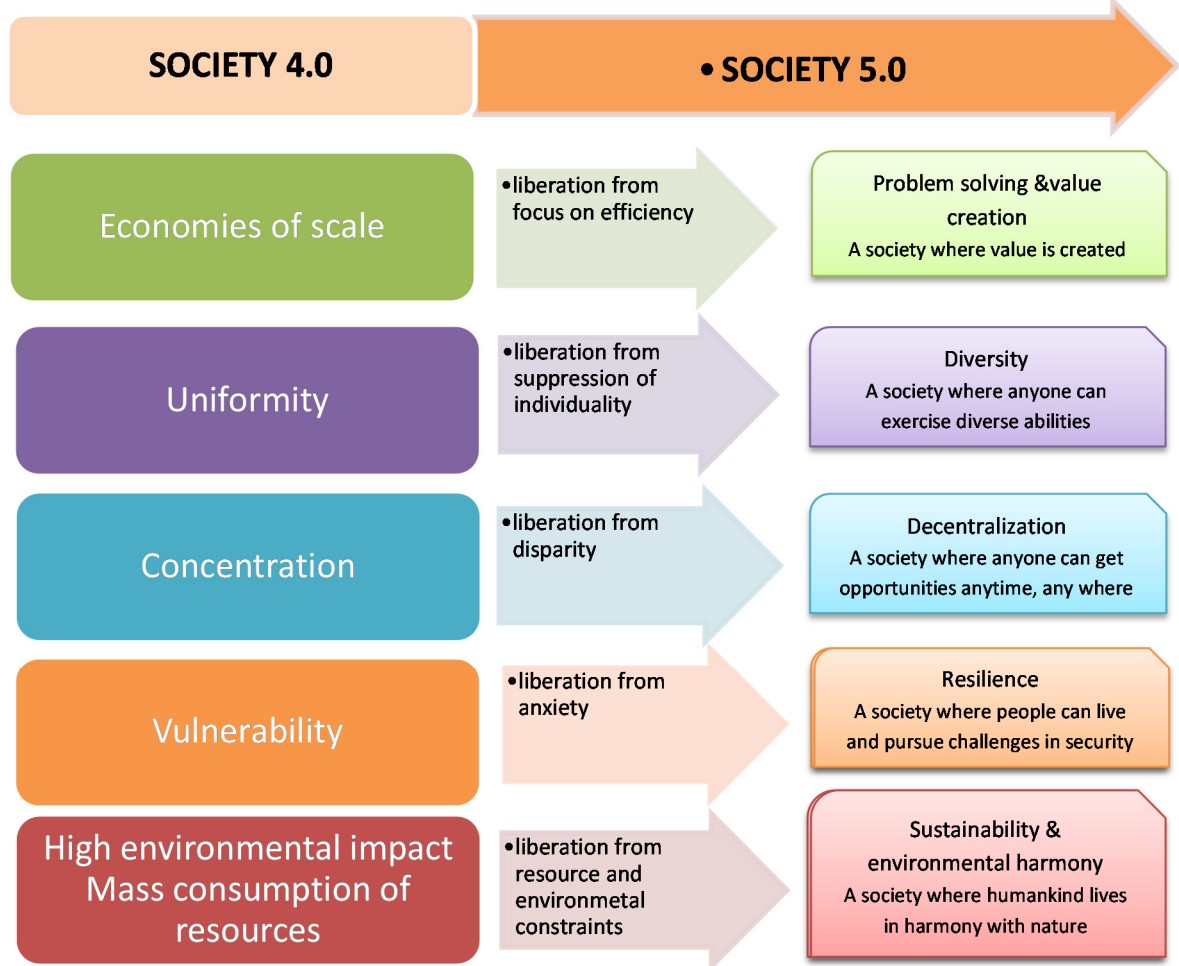

**Figure 1.** Changes from Society 4.0 to Society 5.0. Source: own work based on [33].

The concept of Society 5.0 assumes that workers will be relieved of the continued focus on productivity. Instead, the emphasis will be on meeting individual needs, problem solving, and creating value. Moreover, they will be able to live, learn, and work without fear of alienation because of their values and way of thinking, or attacks of discrimination based on sex, race, nationality, etc. People will be relieved of the imbalances caused by a focus on wealth and information, and everyone will be able to get rich anytime, anywhere. People will be freed from anxiety about terrorism, disasters, and cyber-attacks, and will live in a safe place, with strong protection mechanisms against unemployment and poverty. People will be freed from resources and environmental constraints and will be able to lead a sustainable life in any region [33]. It will all be possible thanks to the use of automatic machines, robots, a new approach to the organisation of work, and the workplace. The new cyber-physical environment, through the implementation of Industry 4.0, will improve connections between people, things, human entities, and technologies in the advanced cyberspace environment. So, in the future, people will need imagination and creativity to change the world, materialise their ideas [34], and also "solve problems" and "create values" that lead to sustainable development.

### 2.3. Social Media in Tourism

Proper communication and the collection, processing, and use of information play an important role in Society 5.0. Nowadays, social media are changing the way the Internet is used and the way we communicate and process information.

In the world, according to the Digital 2020 survey, globally, over 4.5 billion people use the Internet and the number of social media users has already exceeded 3.8 billion [35]. Almost 60% of the world's population is online, and the latest trends suggest that—soon—more than half of the world's population will be using social media. In Poland, every year, we also observe an increase in the use of social media by society. In January 2020, there were 19 million active social media users in Poland, which was 50% of the population [36], compared to the previous year [37], as many as 1 million joined this group. Poles spend an average of 2 h a day using social media, most often on YouTube (92%), while the second most popular site is Facebook (89%). Among social media users, the largest group are the representatives of Generation Y (people aged 25–34) (28%), followed by Generation Z (people aged 18–24) (21%), and Generation X (people aged 35–44) (20%).

Social media are characterised by a high degree of interactivity, using various and widely available types of applications and extensive communication techniques. According to [38] (p. 61), social media (SM) are a group of web applications based on the ideological and technological foundations of Web 2.0, enabling the creation and exchange of user-generated content. Various classifications of social media are available in the literature [39–41], the most frequently cited is the division proposed by [38], who classified them according to two groups of factors:

- social presence/media richness,
- self-presentation and self-disclosure.

Social media focused on building and maintaining relationships, which is the most popular form of social media. Examples include social networks and virtual worlds. In this article, based on [38] as well as [41], the following types of social media have been distinguished:

- blogs and microblogs (e.g., Twitter),
- social networks (e.g., Facebook, Google+),
- professional social networking sites (e.g., LinkedIn),
- cooperation networks/shared projects (e.g., Wikipedia),
- Internet forums (e.g., Globetrotter, Fly4Free, Lonely Planet travel forums),
- content communities (e.g., YouTube, Vimeo, Pinterest)
- rating services and portals (e.g., TripAdvisor, Booking, HolidayCheck),
- virtual worlds, social (e.g., Second Life),
- virtual game worlds (e.g., World of Warcraft).

Initially, social media were mainly used in the area of marketing, [5,6] sales, customer service, or customer relationship management [42]. However, over time, they began to be used for employee recruitment [43–45], internal communication, cooperation [46], project management [47,48], training and education [49], or generating ideas in the so-called open innovation model [50].

Due to the development of new technologies, social media will play an increasingly important role in marketing activities in the field of tourism. Already, SM is used to build the brand of the region, company, and attractions, as well as build relationships with tourists before, during, and after a tourist stay. We can also observe the importance of this communication channel on the basis of spending on marketing activities. In 2019, companies spent 28% of their digital spending on social media—and among the tools, we can see a varied frequency of their use [51]. Therefore, social media, as an effective information channel, should be monitored on an ongoing basis and used even more by marketers (hotels, restaurants, city authorities) in order to effectively reach potential recipients with their promotions.

It should also be noted what an important role in the process of sustainable return to tourism is played by social media, which is used by an increasing amount of the population. Forty-four percent of travellers have increased the time spent browsing through social media during COVID-19 [17]. Tourists are more and more willing to use social media at various stages of their journey, including planning, realising, and sharing travel experiences. Research shows that recreational tourists used an online travel agency (52%), recommendations, such as TripAdvisor (44%), destination-specific (18.5%) [52], when travel planning. In turn, 40.3% of respondents were inspired to travel within their home country through travel information they read on the Internet, while 21.9% of respondents through the opinions of friends and acquaintances in social media [53]. Posts posted by digital influencers have an impact on making decisions about choosing a tourist destination. Research shows that digital influencers' posts influenced 55.1% of respondents to decide on visited destinations, 45.1% of respondents to decide on restaurants or dining, 43.2% of respondents to decide on a hotel, and 26.5% of respondents to decide on personal safety tips [54]. In turn, most respondents indicated that digital influencers had an impact on the choice of place in the form of beach destination or resort (46.6%), national park (41.7%), and large city (39.7%) [54]. The WEX U.S. Travel Trends Report 2019 similarly shows that Instagram is a significant influencer on where younger travellers book trips—with 22% of Millennials and 30% of Gen Z being influenced by the platform [55]. This data indicates that, when looking for a travel destination, the opinions of others posted on social media are an important source of information. Social media make it possible to collect information about the economic and emotional situation of potential customers and reach them quickly and directly. Therefore, keep this in mind when planning your marketing activities.

The public is increasingly posting information and looking for information on social media when travel planning. However, it can also be used to observe their behaviour, collect data that allows tracking whether a given advertisement will influence them. Currently, UNWTO, to recover tourism, recommends creating digital events aimed at a large number of recipients, where social media users have free access to certain parts of the events in order to obtain a base of social media followers [56]. The data collected about these users can be used to create profiles of potential customers, which will allow more effectively reaching them with a promotional message and encourage them to visit a given destination. This indicates that, by collecting information about social media users, it is possible to influence the behaviour of representatives of particular generations.

### 2.4. The Travel Experience of Different Generations

Depending on the age, people planning a trip have different expectations, requirements, or preferences regarding the purpose, place, and time of spending their holiday. In preparing the appropriate promotion of tourist destinations and developing an effective communication strategy, it is important to know how different age groups spend their

free time, their expectations, and needs [13]. Due to the changing economic, business, and political conditions, various age groups of generations can be distinguished over the last several decades. However, this division is not unambiguous in the literature on the subject and differs depending on the country. Nevertheless, assuming that a generation is a group of people who share a similar time of birth and historically, culturally or socially significant events, now it is possible to distinguish four generations of people, whose age allows them to travel [57]. These are Baby Boomers (born in the years 1945–1964), Generation X (born in the years 1965–1980), Generation Y (born in the years 1981–1994), and Generation Z (born after 1995). Unfortunately, there is no unanimity among scientists in defining the age range of generations [58]. This particularly applies to Generation Z. For example, [59] (p. 998) indicates 1990 as the beginning of Generation Z, [60] (p. 110), as well as [60] (p. 110) indicate 1991, while [11] (p. 12), [14] (p. 44), as well as [12] (p. 559) indicate that Generation Z does not start until after 1995.

### 2.4.1. Baby Boomers

Born in the years 1945–1964, they are the so-called Baby Boom (BB), i.e., the generation of the demographic and economic boom. This generation is travelling more and more often and more willingly, relaxing in various places around the world. The increasing mobility of this generation is primarily related to their well-established social position, financial possibilities, and having more free time. The oldest of this generation are already retired, while those, who are still working, need more time to regenerate, relieve stress, and relax. The needs and interests of the Baby Boom generation are changing, as this generation has grown wealthier than in previous years. According to the annual the American Association of Retired Persons (AARP) surveys, the average American Baby Boomer plans to spend USD 7800 on travel in 2020, a significant increase from USD 6600 in 2019 [61]. AARP is the nation's largest nonprofit, nonpartisan organization dedicated to empowering Americans 50 and older to choose how they live as they age. With nearly 38 million members (2018) and offices in every state, the District of Columbia, Puerto Rico, and the U.S. Virgin Islands, AARP works to strengthen communities and advocate for what matters most to families with a focus on health security, financial stability, and personal fulfilment. Half of all Baby Boomer travellers (51%) expect to travel abroad in 2020, taking an average of one or two international trips [62]. At the same time, they are ready to spend more money on their travels, go to expensive places, and are willing to stay longer in higher-quality accommodation than people from Generations X or Y. According to AARP, 85% of BB travellers use the Internet for travel planning [9]. BBs take their smartphones with them when they travel, while it is used by 84% of BBs on foreign trips and 94% on domestic trips. The BB generation primarily uses smartphones for communication, taking pictures, using maps or finding restaurants, and various activities. Most Boomers (80%) save their holiday memories and share them using digital methods, such as texting or sending pictures via text messages (44%), Facebook posts (32%), or digital photo albums (30%) [9] (p. 43). On holiday, people from the Baby Boom generation are more likely to engage in themed trips, as well as personal visits to various types of museums. Moreover, they are looking more for quiet and peaceful places, in contrast to the millennial generation, who more often takes part in gastronomic tours, adventures, theatres, and activities giving a high level of adrenaline. BBs show a great willingness to connect with the local culture and community they visit, as over 50% show interest in local cuisine, traditions, entertainment, and cultural nuances when they are abroad. According to [63], the main motivators of the Baby Boomers generation in choosing holidays are: having fun and enjoying the trip, relieving stress and tension, relaxation, the need for change and novelty, as well as the attractiveness of the physical environment and taking care of better health. They are also considered "active holiday" fans, meaning they prefer activities such as golf, hiking, massage, and wine tasting tours.

### 2.4.2. Generation X

Generation X are people born in the years 1965–1980, who are now 40–55 years old, and who grew up in Poland and entered adult life in times of economic restructuring and political changes. High inflation, increased unemployment, and employment instability forced them to accept unfavourable working conditions and accept jobs below their skills and qualifications. The times, when they had to start their careers, turned them into enterprising people, but also increased the atmosphere of uncertainty and fear of losing their jobs. Similar in the United States, Generation X is described as born in bad economic conditions [64] and their attitudes and beliefs were shaped by the first war in Iraq, the atmosphere surrounding school shootings, reality shows, or the HIV epidemic [65]. Family and friends are very important to this generation [66], therefore, they plan trips together with family and children or friends (54%) [9] (p. 50). Members of this generation appreciate the balance between professional and private life and are eager to look for new places, where they can learn about different customs, culture, and traditions of the local community According to [67], as many as 71% of this generation like to discover new places off the beaten track and look for local recommendations, while 70% visit museums, historical places, as well as art and culture [67]. Therefore, they often visit popular cities, where you can find interesting monuments or rich history. By choosing travelling abroad, they usually look for quiet, recreational places or interesting cultural monuments. Despite the fact that people from this generation were born in analogue times, they fit perfectly in the modern digital world and modern technologies, which is why they willingly use the Internet when planning and booking their tourist trips.

### 2.4.3. Generation Y

Generation Y, also known as Millennials, are people who are currently between 26 and 39 years old. The phenomenon that shaped Generation Y was globalisation, which caused the blurring of boundaries between countries, the merging of cultures, as well as increased availability of products and services from around the world. Double earnings of parents have become a standard, guaranteeing good conditions for the personal development of their children. The main values that guide Millennials are independence, ambitions, creativity, innovation, and development. Respect for ethics, multiculturalism, awareness of social problems, and the possibility of using information and communication technology is of great importance for this generation. This generation is extremely mobile, willing to travel and move from one place to another. This generation has acquaintances and friends not only in their home country but also abroad. Generation Y travels more than Baby Boomers or Generation X. They visit and explore more destinations, spend more while travelling, and are hungry for interesting experiences and information [63,68]. According to [67], Millennials make the largest number of trips per year. They are frequent, but undecided travellers, who enjoy experiencing and discovering nature, often with young children in their arms. This coincides with many other studies of this generation, which indicate that they are confident, ambitious, and achievement-oriented people [11] who prefer to spend money on experiences that enrich their skills, rather than on ordinary things. However, as a group, they do not prefer any particular type of holiday. In fact, Millennials want diversity. Some like experiences, full of adventures and unique experiences [8], while others want to relax on the beach and taste the local cuisine [9] (pp. 52–53). They are also looking for all-inclusive, relaxing, and romantic trips. However, they keep an eye on their travel budget [67]. Research conducted in Poland by [69] indicates that this generation rarely travels alone (less than 7%); they prefer to choose the company of friends (70%) [69].

Moreover, as indicated by the research of [7], as well as [8] people from Generation Y show a strong need to use the opportunities generated by the Internet and social media, both when planning and sharing their experiences during and after the trip. This is because Millennials have reached the age of maturity in the digital world and often think about sharing their experiences—including travel—online. Their young children, Generation Alpha (born after 2010), are completely digital [11]. Each week, 2.5 million

people from the Alpha generation are born worldwide. By 2025, this generation will have 2 billion members. They are expected to be the most transformative generation. Moreover, they already influence the behaviour of their millennial parents in terms of spending money, including planning and spending free time [67]. Moreover, people from Generation Y highly appreciate the comfort of travel and follow various and unique experiences, and not only expect passive rest. According to [70], tourists increasingly avoid mass tourism, wanting to be seen more as travellers rather than tourists [70]. This is related to greater independence in travel planning, as well as their concern for the environment [71]. Compared to Baby Boomers and Generation X, Generation Y is much more active when it comes to planning holidays or booking accommodation via the Internet (dedicated websites, e.g., booking.com), including social media, such as TripAdvisor [72]. Travel organisers looking to reach Millennials can do so by promoting places with fun activities, experiences, and attractions that are of interest to the whole family, as well as making family travel research an interactive, fun, and easy experience for all family members [67].

### 2.4.4. Generation Z

Generation Z consists of people born in the years 1995–2010 and, although they have many features in common with Millennials, there is a consensus among researchers [14,73] that, although some features are more visible in them, they actually differ significantly in many respects. Generation Z is a generation open to the world and novelties, not only technological but also those related to exploring new places. This is a multitasking generation, where social media is their main form of communication. People from Generation Z are very open to the world and willingly undertake various forms of tourist activity, although they often lack financial resources because they are not yet professionally employed and are dependent on their parents. Therefore, due to financial restrictions, domestic recreation—with family or friends—dominates in this group. A survey conducted by Expedia Group Media Solutions in 11 different countries in 2018 revealed that representatives of Generation Z most often travel to rest (54%), visit interesting places (44%), or visit family (42%). The Expedia Group Media Solutions analysis also shows that 84% of Generation Z believe that social media plays an important role when travelling (Generation Y: 77%) and over 50% use platforms such as Twitter, Snapchat, Facebook, Instagram, and YouTube when planning and during the trip [74].

Despite many studies on the use of social media by different generations, there is still a research gap in analysing the generational differences in the way, frequency, and behaviour of generations in the field of sustainable tourism. Our new approach will allow an in-depth analysis of travel planning behaviour by representatives of different generations in order to be able to better adapt promotional campaigns for them. Such research will allow for the preparation of effective promotion using social media to accelerate the return of tourism in the world. Therefore, in order to achieve the aim of the article, the following research questions and hypotheses were asked:

- **RQ1**: What is the frequency of using different social media depending on the generation?

**Hypothesis 1.1.** *There are statistically significant differences between the generations in the frequency of using different types of social media.*

**Hypothesis 1.2.** *Age moderates the strength of the relationship between the frequency of using different types of social media.*

- **RQ2**: What are the behaviours of particular generations in social media in terms of planning a holiday?

**Hypothesis 2.1.** *Recommending a holiday with positive opinions and comments in SM is significantly different for individual generations and gender groups.*

**Hypothesis 2.2.** *Information searched for in SM regarding hindrances that may arise during travel is significantly different for individual generations and gender groups.*

**Hypothesis 2.3.** *Getting to know the history and culture of tourist places using SM is significantly different for individual generations and gender groups.*

**Hypothesis 2.4.** *Resigning from a holiday on the basis of negative opinions and comments in social media is significantly different for individual generations and gender groups.*

**Hypothesis 2.5.** *Planning a trip with the use of social media is significantly different for individual generations and gender groups.*

**Hypothesis 2.6.** *Establishing relationships with the local community through social media is significantly different for individual generations and gender groups.*

**Hypothesis 2.7.** *Checking opinions about tourist places in social media is significantly different for individual generations and gender groups.*

**Hypothesis 2.8.** *Opinion on the use of short-term apartment rentals is significantly different for individual generations and gender groups.*

## 3. Materials and Methods

### 3.1. Measures, Data Collection, and Sample

Research on the use of social media (SM) in tourism was conducted in Poland in the last quarter of 2019 and supplemented in 2020. The research questionnaire consisted of seven questions. The first three questions concerned the characteristics of the respondents' general behaviour in social media. Therefore, the question was asked how long and how often the respondents use SM and what profile settings they have in SM. Responses to these questions were measured on a nominal or ordinal scale. The four remaining questions had an extensive structure, including the study of respondents' agreement with the opinions on the use of social media in tourism. They were asked about their opinions on the use of SM at the stage of planning the trip, and their behaviours during this process. Responses to these questions were measured using the Likert scale. Thanks to the Likert scale used, it was possible to choose one of five response variants, arranged symmetrically, in terms of positive or negative reference to the issue being addressed. The choice of an intermediate answer meant a neutral attitude to the issue under review or no opinion on a given topic. The obtained answers were coded in such a way that a positive attitude to a given phenomenon was graded with a value of 2—for a strongly positive answer, or 1—when the positive answer was not supported only partially. In the case of a negative grade, the assigned value was −2, unless it was partially negative, then the assigned value was −1. The neutral grade was awarded a value of 0. With such coding of the responses, the mean value of a given variant of the response higher than zero means a positive attitude towards the issue by all respondents, whereas a negative mean value indicated a negative attitude to given opinions of all respondents. This made it possible to compare the responses given by nonparametric statistical tests in individual groups designated by generational identification. The main questions were followed by a record specifying their gender, age, and education, as well as tourist experience, differentiated by the length and frequency of tourist trips.

After the initial selection of collected questionnaires, 397 respondents were qualified for further analysis, which exceeded the minimum random sample size estimated at 386 questionnaires (for the assumed maximum statistical error rate of the sample of ±5%

and the confidence level $p$ = 0.95). Table 1 contains detailed information on the respondents participating in the survey.

**Table 1.** Structure of respondents.

| Generations | [%] | Gender of Respondents [%] | [%] |
|---|---|---|---|
| Baby Boom (BB) | 2.00% | Female | 59.4% |
| X | 19.60% | Male | 40.6% |
| Y | 21.40% | | |
| Z | 57.00% | | |
| **Level of education** | **[%]** | **Length of using SM** | **[%]** |
| Basic/Junior high | 1.0% | Under 6 years | 78.84% |
| Secondary | 38.6% | From 4 to 6 years | 8.06% |
| Higher I | 27.5% | From 2 to 4 years | 2.52% |
| Higher II | 30.8% | Up to 2 years | 2.27% |
| Postgraduate | 1.0% | I don't remember | 8.31% |
| PhD | 1.0% | Never used | 0% |

Source: based on own study.

When analyzing the sample structure, a small percentage of the BB generation participants can be seen. The number of responses from the BB generation was limited due to their limited use of new technologies. However, despite the small size of this subgroup of respondents, we decided to include it in the comparisons being at the center of our research problems. Therefore, we formulate all our conclusions bearing in mind the limitations resulting from the low presence of representatives of the BB generation.

*3.2. Data Analysis*

The processing of the collected data consisted in performing statistical analyses—one-dimensional, two-dimensional, and three-dimensional. One-dimensional analysis was performed based on classical and positional descriptive analysis. Pearson's correlation coefficients, Spearman's rank correlation, and Kendall's tau were used in the two-dimensional analysis. The level of significance of differences between the obtained mean values for individual generations and gender groups was also examined. The Mann–Whitney U test was used to compare two groups with a non-normal distribution, and the Kruskal–Wallis test was used for many groups of variables with a non-normal distribution. In the case of detection of a relationship between the variables, further analyses were undertaken to explain the nature of this relationship, using correspondence analysis and association analysis. In the course of three-dimensional analysis, when analysing the dependencies of pairs of features, it was also examined whether these relationships are moderated by generation, i.e., whether the generation variable interacts with individual predictors. For this purpose, the significance of the interaction effect was examined during the regression analysis (according to the methodology of [75]. All statistical tests were performed considering significance at the level of $\alpha$ = 0.05

**4. Results and Discussion**

*4.1. Activity of Tourists of Various Generations in Social Media*

Answering the first research question, RQ1, the above-mentioned frequencies were analysed and the Chi-square test of independence, the Kruskal–Wallis test and correspondence analysis were performed. Correlation coefficients between individual pairs of social media were also examined, as well as regression and association analyses were performed. The conducted analyses were to verify the research hypotheses: Hypotheses 1.1 and 1.2.

4.1.1. The Frequency of Using Social Media Depending on the Generations

An analysis of the frequency of using social media has shown that Generations X, Y, and Z have been using SM for over four years or do not remember how long they have

been using them, while BBs are a generation that has been using SM for a short time (up to two years).

The BB generation does not use or occasionally uses most social media. The exceptions in this regard are social networks, such as Facebook, which 28.6% of this generation use several times a week, and the same number a few times a day. Cooperation networks/shared projects are also slightly less popular in this generation, as they are visited once or several times a week or a few times a day by 28.6% of respondents.

Generation X does not use blogs and microblogs (42%) or uses them occasionally (43%). They use social networking sites a few times a day (40%), although there are also people who use them all the time (15%) and those who only use them a few times a week (14%). Another medium often used by Generation X is content communities (e.g., YouTube, Vimeo, Pinterest)—18% of the respondents use it a few times a day, 14% once a day, and 32% several times a week. Generation X mostly uses cooperation networks and shared projects, as well as occasionally rating portals, although there are those who use it several times a week or once a week (14–16%). Generation X rarely uses professional social networking sites (LinkedIn, GoldenLine): 11% use it several times a week and 10% once a day. Travel forums are underused or used occasionally, and around 13% use them once a week.

The manner and frequency of using media such as microblogs, travel forums, and rating portals for Generation Y is similar to Generation X, and so they never use it or do it occasionally. Generation Y uses social networking and content services much more often than Generation X. More than half of the representatives of Generation Y do it all the time, and 1/3 of them a few times a day. The higher frequency of use among Generation Y compared to Generation X is also visible in the content media. One third of them uses it a few times a day and 10% all the time. These frequencies are—therefore—twice as high as in the case of Generation X. Generation Y is slightly less active in professional social networking sites than in the case of Generation X.

Generation Z is a generation that primarily uses social networks, as 90% of respondents use it all the time or a few times a day. Content services are also very popular: almost half of Generation Z respondents use them a few times a day, and about 1/3 all the time. These indicators are, by far, the greatest compared to all other generations.

An aggregated approach to the frequency of using individual types of social media is presented in Table 2.

**Table 2.** Descriptive statistics on social media usage frequency for generations.

| | BB | | | X | | | Y | | | Z | | |
|---|---|---|---|---|---|---|---|---|---|---|---|---|
| | **Mean** | **Median** | **Sd** | **Mean** | **Median** | **Sd** | **Mean** | **Median** | **Sd** | **Mean** | **Median** | **Sd** |
| Blogs and microblogs | 0.43 | 0.00 | 0.53 | 1.05 | 1.00 | 1.56 | 0.86 | 1.00 | 1.29 | 1.15 | 1.00 | 1.59 |
| Social portals | 2.71 | 3.00 | 1.89 | 3.99 | 5.00 | 1.71 | 4.93 | 5.00 | 1.25 | 5.34 | 5.00 | 0.74 |
| Cooperation networks | 1.14 | 0.00 | 2.27 | 1.88 | 1.00 | 1.50 | 2.10 | 2.00 | 1.48 | 2.47 | 3.00 | 1.34 |
| Content communities | 0.71 | 1.00 | 0.49 | 3.16 | 3.00 | 1.52 | 3.94 | 4.00 | 1.43 | 4.78 | 5.00 | 1.17 |
| Rating portals | 0.43 | 0.00 | 0.53 | 1.58 | 1.00 | 1.31 | 1.51 | 1.00 | 1.26 | 1.22 | 1.00 | 1.18 |
| Travel forums | 0.71 | 1.00 | 0.49 | 0.92 | 1.00 | 1.00 | 1.12 | 1.00 | 1.22 | 0.80 | 1.00 | 1.18 |

Scale: 0: not at all, 1: occasionally, 2: once a week, 3: a few times a week, 4: once a day, 5: a few times a day, 6: all the time. Source: based on own study.

The Chi-square test of independence showed no significant statistical dependence on the generation frequency of using blogs and microblogs. In turn, the Kuruskal–Wallis test also showed no dependence of the generation on the frequency of using travel forums. In the case of other social media, the tests show that the frequency of use was dependent on the generation. Therefore, the conducted research shows that the Hypothesis 1.1 research

hypothesis was partially confirmed. However, we formulate this conclusion taking into account the limitations of our research resulting from the small number of representatives of the BB generation in the research sample.

By examining the nature of the emerged dependencies, tests of multiple comparisons of all ranks were carried out. In the case of social networks, these tests showed the following differences: ZX ($p < 0.0000001$), ZBB ($p = 0.0009$), YX ($p = 0.003$), YBB ($p = 0.02$). Generations Z and Y use social networks more often than Generations X and BB. The graphical presentation of the dependence of the frequency of using social networking sites from generation to generation was made on the basis of correspondence analysis (Figure 2).

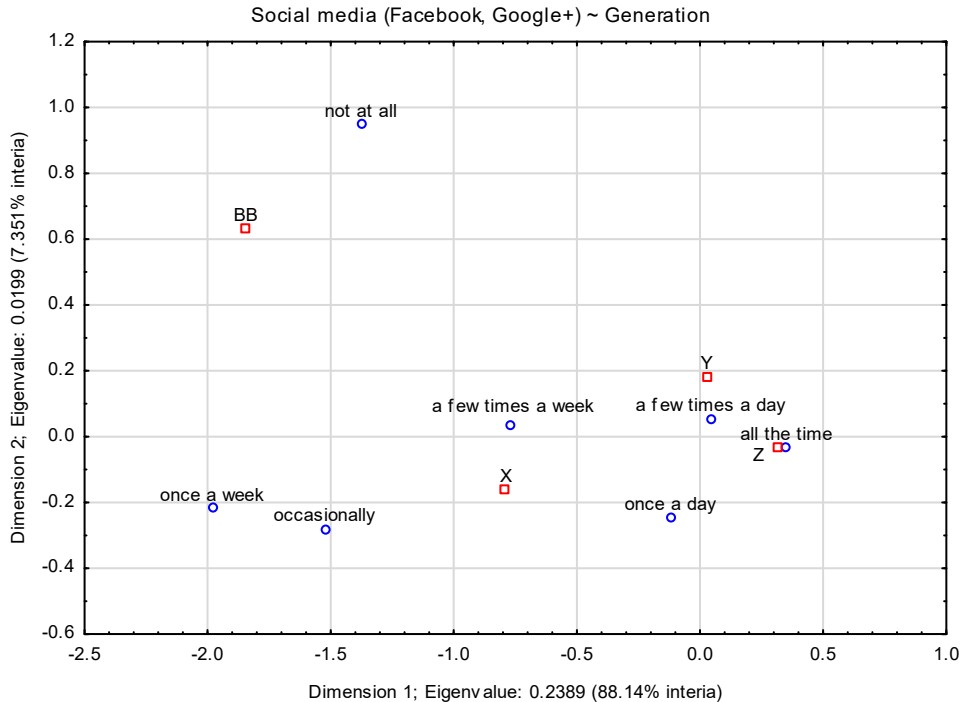

**Figure 2.** Correspondence analysis for the variables "Social networks" and "Generation". Source: own study.

Dimension 1 clearly differentiates Generations Y and Z from Generations X and BB. Generations BB and X appear closer to the "once a week" value in the chart. Generations Y and Z are closer to the "all the time" value.

The Chi square test of independence and the Kruskal–Wallis test was performed to examine the dependence of the frequency of using particular social media on the generation and gender. The results are summarized in Table 3.

**Table 3.** The results of the study of the dependence of the frequency of using social media from generations.

| Types of Social Media | Generation | | | |
|---|---|---|---|---|
| | Chi Kwadrat | *p*-Value | Kruskal-Wallis | *p*-Value |
| V1.1. Blogs and microblogs | 8.48 | 0.970 | 1.85 | 0.61 |
| V1.2. Social portals | 107.34 | <0.0001 | 62.49 | <0.00001 |
| V1.3. Cooperation networks | 87.47 | <0.0001 | 20.79 | 0.0001 |
| V1.4. Content communities | 180.67 | <0.0001 | 91.43 | <0.0001 |
| V1.5. Rating portals | 31.31 | 0.0018 | 15.04 | 0.018 |
| V1.6. Travel forums | 22.82 | 0.029 | 7.64 | 0.05 |

Source: own study.

In the case of content communities, there were differences between all pairs of generations ($p < 0.05$), except for the XBB pair. Generations Y and Z are characterised by a greater frequency of activity in content communities. Correspondence analysis shows that the differentiation of generations is most influenced by the value of "not at all", which is mainly the focus of the BB generation (Figure 3).

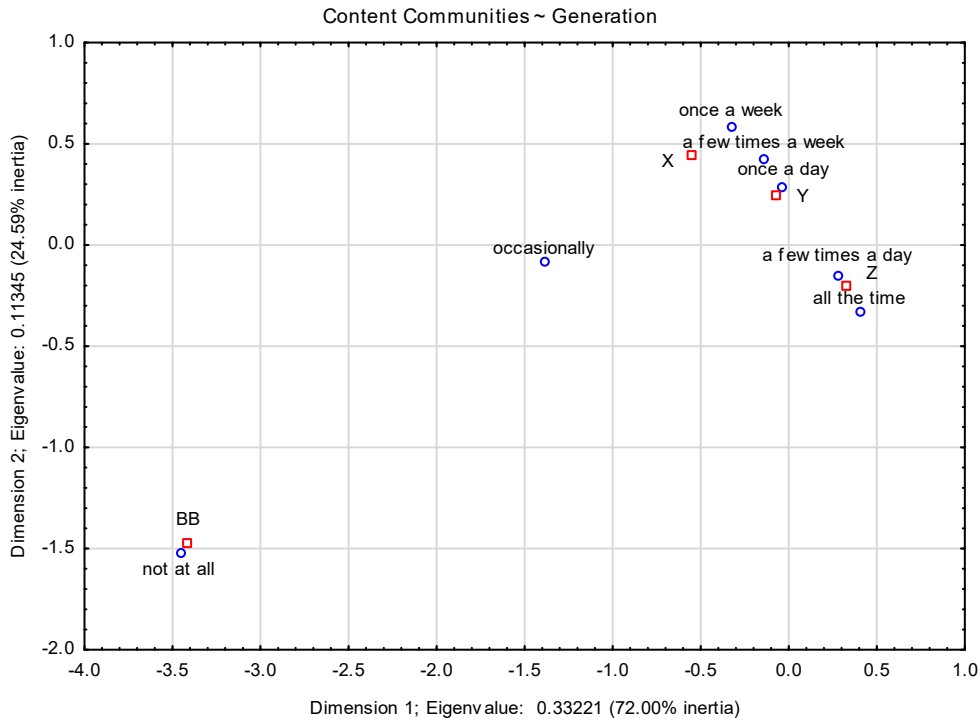

**Figure 3.** Correspondence analysis for the variables "Content communities" and "Generation". Source: own study.

By examining subsequent social media, it was found that, in the case of cooperation networks, statistically significant differences exist for the pairs: ZX ($p = 0.004$) and ZBB ($p = 0.03$), which means that Generation Z uses cooperation networks more often than Generations X and BB. The correspondence analysis showed that the value having the greatest impact on the differentiation of generations was the value of "not at all", which was mainly shown by the representatives of the BB generation. In the case of rating portals, there were differences for all pairs ($p < 0.05$) except for the XY pair. The correspondence analysis revealed that the value "a few times a day" had the strongest impact on the differentiation of generations.

To summarise, we can observe a different frequency—depending on the generation—of using the following types of social media, i.e., social portals, cooperation networks, content communities, and rating portals. Only the frequency of using blogs and microblogs, as well as travel forums did not differ depending on the generation. This is confirmed by research conducted by [35,37].

### 4.1.2. Relationships between Different Types of Social Media

To examine the relationship between the types of social media, Spearman's rank correlation coefficient and Kendall's tau were calculated. They were gathered in Tables 4 and 5.

**Table 4.** Spearman's rank correlation coefficients (all generations combined).

| Types of Social Media | V1.1 | V1.2 | V1.3 | V1.4 | V1.5 | V1.6 |
|---|---|---|---|---|---|---|
| V1.1. Blogs and microblogs (e.g., Twitter) | 1.000 | 0.023 | 0.166 | 0.073 | 0.110 | 0.185 |
| V1.2. Social portals (e.g., Facebook, Google+) | 0.023 | 1.000 | 0.243 | 0.461 | 0.164 | 0.083 |
| V1.3. Cooperation networks (e.g., Wikipedia) | 0.166 | 0.243 | 1.000 | 0.270 | 0.155 | 0.105 |
| V1.4. Content communities (e.g., Youtube, Vimeo, Pinterest) | 0.073 | 0.461 | 0.270 | 1.000 | 0.120 | 0.035 |
| V1.5. Rating portals (e.g., TripAdvisor, Booking, HolidayCheck) | 0.110 | 0.164 | 0.155 | 0.120 | 1.000 | 0.514 |
| V1.6. Travel forums (e.g., Globetrotter, Fly4Free, Lonely Planet)) | 0.185 | 0.083 | 0.105 | 0.035 | 0.514 | 1.000 |

Source: own study.

**Table 5.** Kendall tau correlation coefficients (all generations combined).

| Types od Social Media | V1.1 | V1.2 | V1.3 | V1.4 | V1.5 | V1.6 |
|---|---|---|---|---|---|---|
| V1.1. Blogs and mikroblogs (e.g., Twitter) | 1.000 | 0.020 | 0.142 | 0.061 | 0.097 | 0.165 |
| V1.2. Social portals (e.g., Facebook, Google+) | 0.020 | 1.000 | 0.207 | 0.405 | 0.143 | 0.075 |
| V1.3. Cooperation networks (e.g Wikipedia) | 0.142 | 0.207 | 1.000 | 0.228 | 0.132 | 0.090 |
| V1.4. Content communities (e.g., Youtube, Vimeo, Pinterest) | 0.061 | 0.405 | 0.228 | 1.000 | 0.102 | 0.031 |
| V1.5. Rating portals (e.g., TripAdvisor, Booking, HolidayCheck) | 0.097 | 0.143 | 0.132 | 0.102 | 1.000 | 0.477 |
| V1.6. Travel forums (e.g., Globetrotter, Fly4Free, Lonely Planet) | 0.165 | 0.075 | 0.090 | 0.031 | 0.477 | 1.000 |

Source: own study.

The study of the dependencies of using the types of social media showed the existence of significant, although mostly not high, dependencies. The highest value of Spearman's rank correlation coefficient and Kendall's tau was recorded for the pair of Travel Forums and Rating Portals (Spearman's correlation coefficient = 0.51, Kendall's tau = 0.48). The second was the pair of Social networks and Content communities (Spearman's correlation coefficient = 0.46, Kendall's tau = 0.41). In order to examine the nature of these dependencies between these two social media pairs, an association analysis was performed. For the pair of Travel forums and Rating portals, the strongest rule was the following:

*Travel forums = I don't use them at all => Rating portals = I use them occasionally (support = 21.4%, trust = 46.2%)*

This means that, in 1/5 of the respondents, a low frequency of using both types of media was observed, and nearly half of all those who did not use travel forums used rating portals occasionally.

The situation is different in the case of the association analysis made for the pair of Social networks and Content communities. The strongest rule here is the following:

*Social networks = a few times a day => Content communities = all the time (support = 18.4%, trust = 44%)*

This means that, in 18% of respondents, a high frequency of using both types of media was observed. However, 44% of all those who used social networking sites a few times a day use content communities all the time.

The correlation between all social media pairs was also examined in subgeneration groups. Similar values of the examined coefficients were observed for Generations X, Y, and Z as for the whole group of respondents. However, for the BB generation, they turned out to be insignificant and different from zero ($p < 0.05$). This means that, for Generations X, Y, and Z, the dependencies in using particular types of social media are weak or average. However, for the BB generation, no such relationships were observed. This may suggest that belonging to a generation is a variable moderating the strength of the relationship between the frequencies of using different types of social media. To verify this, a regression analysis was carried out with the inclusion of a predictor resulting from the interaction of a given social medium with the variable generation, and it was checked whether the β coefficient was statistically significant. The obtained β coefficients were statistically significant only in two cases:

1.   for V1.3 and V1.5 ($p$ = 0.00043, the correlation coefficient for Generations X and BB is not significantly different from zero, for Generation Y 0.33 and for Z 0.22)
2.   for V1.4 and V1.2 ($p$ < 0.00001, the correlation coefficient for the BB generation is not significantly different from zero, for X 0.32, for Y 0.42, for Z 0.24).

This means that the Hypothesis 1.2 hypothesis should only be accepted partially. However, we formulate this conclusion taking into account the limitations of our research resulting from the small number of representatives of the BB generation in the research sample.

### 4.2. Using SM at the Planning Stage of the Trip

Answering the second research question, RQ2, nonparametric tests were performed to compare multiple independent samples. The performed analyses were to verify the research hypotheses: Hypotheses 2.1–2.8.

Analysing the overall answers of the respondents (without the division of generations) to the questions asked about their agreement with the opinions presented to them, one can indicate opinions, which were perceived as having a high degree of identification with them and which the respondents did not agree with (Figure 4). These opinions can be ranked according to the decreasing degree of agreement of the respondents. This is the following order:

- I check opinions/stories on places I want to visit on social media ($1.2 \pm 0.84$).
- Positive opinions and comments in social media encourage me to go on holiday ($1.12 \pm 0.9$).
- In social media, I am looking for information about hindrances and problems that may arise in the places I intend to visit ($1.06 \pm 0.81$).
- I use social media to learn about the history and culture of tourist places ($0.94 \pm 0.93$).
- I use social media to plan a trip ($0.82 \pm 0.97$).
- Negative opinions and comments in social media make me resign from a holiday ($0.52 \pm 1.01$).
- I use short term apartment rentals (e.g., Airbnb) ($-0.07 \pm 1.4$).
- I use social media to establish relationships with the local community ($-0.17 \pm 1.26$).

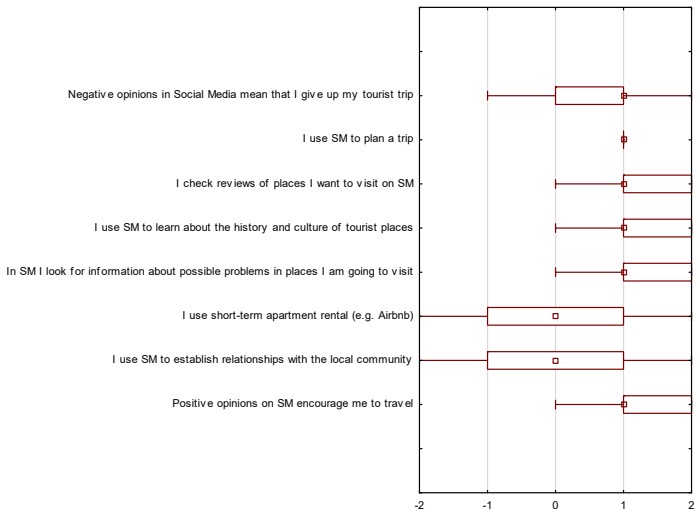

**Figure 4.** Social Media (SM) in planning a trip. Source: own study. $-2$: I strongly disagree, $-1$: I rather disagree, 0: I don't have an opinion, 1: I rather agree, 2: I strongly agree.

The presented research results are worth discussing in the context of the results obtained by [76]. They point out that Facebook posts reinforce the intention to visit a tourist destination and explain the dependence of these variables with the mediating variable in the form of benign envy. High agreement of the respondents in our research with

the statements "Positive opinions and comments in social media encourage me to go on holiday" and "I check reviews on places I want to visit on social media" can indicate the occurrence of the phenomenon mentioned by [77]. It should also be noted that the high agreement with the indicated statements is in line with [76,78] conclusions concerning the role of SM in destination branding. While discussing the results shown in Figure 4, it should be added that such a large role of SM in travel planning is conducive to the creation of applications supporting education in the field of history and culture of a given tourist destination. An example of this kind of application can be found at work [79].

Generational and Gender Influences

In accordance with the adopted methodology for assessing individual issues, the following generational and gender differences can be noticed (see Tables 6 and 7):

- In case of the statement that positive opinions and comments on social media encourage to go on holidays, Generation Z and BB agreed with it more than Generation X and Y. Comparative analysis between generations confirmed that there is a statistically significant difference between the mean scores of Generations Z and X ($p = 0.013$), as well as Z and Y ($p = 0.033$). The gender analysis showed significant differences between the opinions of women and men ($p = 0.001$), where women expressed their positive agreement with the discussed view more often than men. In this way, the Hypothesis 2.1 was confirmed.
- Respondents of all generations agree that they use SM to learn about the history of these places and look for information about hindrances in the places they plan to go to on holiday. There were no generational or gender differences. Thus, the Hypotheses 2.2 and 2.3 were rejected.
- In turn, in the case of the role of negative comments, Generation Z declares their greatest share in the respondent's resignation. Comparative analysis between the generations confirmed that there is a statistically significant difference between the mean scores for Generations Z and Y ($p = 0.006$). The gender analysis did not show any statistically significant differences. Thus, the Hypothesis 2.4 was partially accepted.
- Generally, all respondents of Generations X, Y, and Z use SM to plan their trip, while Generation BB maintains a neutral attitude in this respect. The Kruskal–Wallis test did not show statistically significant differences here. Women more often than men declare a positive attitude towards using SM to plan a trip. A statistically significant difference was observed in this respect ($p = 0.001$). Thus, the Hypothesis 2.5 was partially accepted.
- Respondents of all generations do not use SM to establish relationships with the local community, and representatives of the BB generation are more assertive in this opinion. There were no statistically significant differences, neither at the generational nor gender level. Thus, the Hypothesis 2.6 was rejected.
- In general, all respondents agree with the opinion that they check opinions/stories on the places they want to visit on social media. The BB generation agrees with this statement much less frequently than the other generations. Comparative analysis between the generations confirmed that there is a statistically significant difference between the mean scores for Generations Z and BB ($p = 0.03$). In turn, a comparative analysis by gender revealed that women much more often than men check the opinions about places they want to visit in SM ($p = 0.002$). Thus, the Hypothesis 2.7 was accepted.
- The greatest support for the use of short-term apartment rentals is declared by Generation X. Generations Y and Z maintain a neutral attitude, and generation BB—negative. Comparative analysis between generations confirmed that there is a statistically significant difference between the mean scores of Generations X and BB ($p = 0.002$), Y and BB ($p = 0.007$), as well as Z and BB ($p = 0.005$). Men more often than women declared a positive opinion on the use of short-term apartment rentals, but there were no statistically significant differences between these groups. Thus, the Hypothesis 2.8 was partially confirmed.

**Table 6.** Results of mean scores of agreement of the surveyed generations with the presented opinions on the use of SM at the planning stage of a trip.

| Statement | Measure | BB | X | Y | Z | K–W Test | Man | Woman | M–W Test |
|---|---|---|---|---|---|---|---|---|---|
| Positive opinions and comments in social media encourage me to go on holiday | Mean<br>Med.<br>Sd | 1.14<br>1.00<br>0.38 | 0.97<br>1.00<br>0.89 | 0.95<br>1.00<br>1.02 | 1.22<br>1.00<br>0.85 | 0.03 | 0.94<br>1.00<br>0.96 | 1.24<br>1.00<br>0.84 | 0.001 |
| In social media, I am looking for information about hindrances and problems that may arise in the places I intend to visit | Mean<br>Med.<br>Sd | 1.14<br>1.00<br>0.38 | 0.99<br>1.00<br>0.82 | 0.95<br>1.00<br>0.90 | 1.12<br>1.00<br>0.78 | 0.34 | 0.98<br>1.00<br>0.79 | 1.11<br>1.00<br>0.82 | 0.063 |
| I use social media to learn about the history and culture of tourist places | Mean<br>Med.<br>Sd | 0.43<br>1.00<br>1.13 | 0.90<br>1.00<br>0.92 | 0.96<br>1.00<br>1.02 | 0.97<br>1.00<br>0.90 | 0.44 | 0.86<br>1.00<br>0.97 | 1.00<br>1.00<br>0.91 | 0.126 |
| Negative opinions and comments in social media make me resign from a holiday | Mean<br>Med.<br>Sd | 0.00<br>0.00<br>1.00 | 0.45<br>1.00<br>1.01 | 0.31<br>1.00<br>1.06 | 0.66<br>1.00<br>0.98 | 0.016 | 0.47<br>1.00<br>1.04 | 0.57<br>1.00<br>0.99 | 0.429 |
| I use social media to plan a trip | Mean<br>Med.<br>Sd | 0.00<br>0.00<br>1.00 | 0.82<br>1.00<br>0.85 | 0.86<br>1.00<br>0.97 | 0.82<br>1.00<br>1.01 | 0.14 | 0.62<br>1.00<br>1.00 | 0.95<br>1.00<br>0.93 | 0.000 |
| I use social media to establish relationships with the local community | Mean<br>Med.<br>Sd | −0.71<br>−2.00<br>1.60 | −0.26<br>0.00<br>1.23 | −0.15<br>0.00<br>1.33 | −0.12<br>0.00<br>1.23 | 0.62 | −0.20<br>0.00<br>1.25 | −0.15<br>0.00<br>1.26 | 0.664 |
| I check opinions/stories on places I want to visit on social media | Mean<br>Med.<br>Sd | 0.71<br>1.00<br>0.76 | 1.10<br>1.00<br>0.87 | 1.11<br>1.00<br>0.87 | 1.28<br>1.00<br>0.82 | 0.04 | 1.05<br>1.00<br>0.89 | 1.30<br>1.00<br>0.79 | 0.002 |
| I use short term apartment rentals (e.g., Airbnb) | Mean<br>Med.<br>Sd | −1.57<br>−2.00<br>1.13 | 0.21<br>0.00<br>1.37 | −0.11<br>0.00<br>1.45 | −0.10<br>0.00<br>1.37 | 0.01 | 0.08<br>0.00<br>1.46 | −0.17<br>0.00<br>1.35 | 0.087 |

Source: own study.

**Table 7.** Results of the nonparametric test of the significance of differences between the mean scores of agreement with opinions in individual generations and gender groups.

| Statement | Level of Significance of Differences between Generations or Gender | | | | | | |
|---|---|---|---|---|---|---|---|
| | ZBB | ZX | ZY | YBB | YX | BBX | MW |
| Negative opinions and comments in social media make me resign from a holiday | 0.079 | 0.119 | 0.006 | 0.434 | 0.361 | 0.241 | 0.429 |
| I check opinions/stories on places I want to visit on social media | 0.030 | 0.072 | 0.088 | 0.133 | 0.914 | 0.141 | 0.002 |
| I use short term apartment rentals (e.g., Airbnb) | 0.005 | 0.094 | 0.960 | 0.007 | 0.164 | 0.002 | 0.087 |
| Positive opinions and comments in social media encourage me to go on holiday | 0.415 | 0.013 | 0.033 | 0.974 | 0.784 | 0.819 | 0.001 |

Source: own study.

However, we form these conclusions taking into account the limitations of our research resulting from the small number of representatives of the BB generation in the research sample.

The obtained research results are not fully confirmed in the AARP studies [9] (p. 43), as different results were obtained in relation to the BB generation. In the study, the BB generation does not use or occasionally uses most social media when travel planning. In contrast, in the AARP study, 85% of BB generation travellers use the Internet for travel planning, including saving their holiday memories with Facebook posts (32%). Moreover, in the study, generation BB does not use social media to establish relationships with the local community, unlike other studies [9]. One can, therefore, presume that this situation is typical of Polish society.

Table 8 summarises the verification of the research hypotheses presented in the study. Results confirmed literature findings on the potential benefits of using social media in marketing and promoting [5,6,56]. The present research additionally shows that the expectations and preferences of each generation group regarding the use of social media differ. Therefore, the collected information on the behaviour of tourists in social media allows for more accurate creation of promotional content. Messages posted on social media can come from a variety of sources and may contain diverse information about a particular location. Our findings are similar to the results of [76] where social media is used for the purpose to promote a destination's image by official organisations, and then of the most popular hashtags related to them, showing user-generated content from the point of view of both destination managers and tourist. Thanks to posts from various sources, the possibility of influencing people planning a tourist trip with different expectations and needs is increased.

**Table 8.** Summary of the hypothesis verification.

| | Hypothesis | Verification Result |
|---|---|---|
| Hypothesis 1.1 | There are statistically significant differences between the generations in the frequency of using different types of social media | Partially confirmed, no significant differences in terms of generation, only for blogs |
| Hypothesis 1.2 | Age moderates the strength of the relationship between the frequency of using different types of social media | Partially confirmed, interaction effect for cooperation networks and rating portals, as well as for content communities and social networks |
| Hypothesis 2.1 | Recommending a holiday with positive opinions and comments in SM is significantly different for individual generations and gender groups | Confirmed |
| Hypothesis 2.2 | Information searched for in SM regarding hindrances that may arise during travel is significantly different for individual generations and gender groups | Rejected |
| Hypothesis 2.3 | Getting to know the history and culture of tourist places using SM is significantly different for individual generations and gender groups | Rejected |
| Hypothesis 2.4 | Resigning from a holiday on the basis of negative opinions and comments in social media is significantly different for individual generations and gender groups | Partially confirmed, no significant differences in terms of generation |
| Hypothesis 2.5 | Planning a trip with the use of social media is significantly different for individual generations and gender groups | Partially confirmed, significant differences in terms of age |
| Hypothesis 2.6 | Establishing relationships with the local community through social media is significantly different for individual generations and gender groups | Rejected |
| Hypothesis 2.7 | Checking opinions about tourist places in social media is significantly different for individual generations and gender groups | Confirmed |
| Hypothesis 2.8 | Opinion on the use of short-term apartment rentals is significantly different for individual generations and gender groups | Partially confirmed, no significant differences in terms of generation |

Source: own study.

According to the results, social portals (e.g., Facebook, Google+) were the most visited by all generations. Therefore, it is worth using these channels in the implementation of marketing activities. It is recommended that travel managers and advertisers should develop marketing [76] and communication strategies considering friends' communications on Facebook as a factor that can influence travel consumers' behaviour and decision making. This is especially important during restart tourism [10] when it will be necessary to reduce

the concentration of tourists in one area. Social media can be used for analysing the spatial concentration of the images captured by Instagram users to identify the most visited locations as shown in [78].

## 5. Conclusions, Limits of Research, and Future Research

In response to the challenges of the modern world in terms of the need to implement sustainable development and the sustainable recovery of tourism, it is necessary to use IT solutions that will allow influencing the behaviour of tourists. The development of Industry 4.0 means that companies should use new IT tools that use data from various sources. Social media is increasingly used to collect and gather information provided by potential and current customers of tourist services. They are increasingly used by both marketers and large amounts of the population. It is possible to influence tourists, including their emotions, through social media. Therefore, it is important to include social media in the development of a marketing strategy, the content of which would encourage people to visit a given place. This can provide an incentive for Society 5.0 that will make widespread use of social media in everyday life. This is especially true of Generations Y and Z, which, unlike their predecessors, cannot imagine functioning in a society without social media. Moreover, by monitoring activity in social media, it is possible to learn about the preferences and behaviours of individual generations at the stage of planning a tourist trip. Acquiring such information can be used to create innovative solutions that will help the sustainability of tourism.

The research conducted by the authors made it possible to obtain answers to the research questions posed in the study. When answering the first research question about the frequency of using social media depending on the generation, it was observed that the frequency of using various types of social media varies significantly depending on the generation. The BB generation does not use or occasionally uses most social media. It declared that it uses Facebook a few times a week, and also occasionally uses content communities and travel forums. In turn, representatives of Generation X use social portals a few times a day and content communities a few times a week. Moreover, they declared that they occasionally use blogs and microblogs, cooperation networks, rating portals, and travel forums. The representatives of Generation Y declared that they use social portals a few times a day, content communities once a day, and cooperation networks once a week. Moreover, they indicated that they occasionally use blogs and microblogs, rating portals, and travel forums. While Generation Z uses social portals and content communities a few times a day, and cooperation networks a few times a week. Like previous generations, they occasionally use blogs and microblogs, rating portals, and travel forums. The obtained results also made it possible to answer the second research question: what kind of behaviour in social media is characteristic of particular generations in terms of planning a holiday. The conducted research shows that, at the stage of planning a holiday, the behaviours observed in SM, which do not differ depending on the generation, are those aimed at achieving the following goals: (1) planning a trip, (2) getting to know the history and culture of tourist places, (3) looking for information about hindrances in places intended for tourism, and (4) establishing relationships with the local community. However, the behaviours in SM at the planning stage of the trip, for which intergenerational differences were observed, are those aimed at achieving the following goals: (1) checking opinions and stories on places intended for tourism, (2) making decisions based on negative and positive comments about a tourist destination, and (3) using short term apartment rentals. The analysis of the obtained research allows for the following conclusions on the subject of these behaviours of individual generations.

- The BB generation differs significantly from Generation Z in terms of checking the opinion of the place they want to visit in SM and the use of short-term apartment rentals (e.g., Airbnb), as they declare these behaviours less frequently than Generation Z. Both negative and positive comments in SM about tourist destinations are treated by the BB generation similarly as i case of Generations X, Y, and Z.

- Generation X differs significantly from Generation Z only in terms of behaviour related to making decisions based on positive comments, as it shows less enthusiasm in this respect than Generation Z. Significant differences were also noted in the field of short-term apartment rentals by Generation X, compared to BB, where Generation X declares a more favourable attitude to this behaviour. In the case of other behaviours, Generation X does not differ from the examined generations.

- In terms of behaviour related to making decisions based on positive and negative comments, Generation Y clearly differs from Generation Z. In both of these behaviours, a more neutral attitude is declared than in case of Generation Z, for which these comments are more important. Representatives of this generation also declare the use of short-term apartment rentals (e.g., Airbnb) the most. This behaviour significantly distinguishes the representatives of Generation Y from the representatives of Generations BB and Z.

- Generation Z is most in favour of making decisions based on positive comments, and in this respect, it differs from Generations X and Y but does not differ from generation BB. In the case of negative comments, only Generation Y declares a more critical attitude than Generation Z. In terms of the use of short-term apartment rentals and checking the opinion of tourist destinations in SM, Generation Z is characterised by a neutral attitude, similar to Generations X and Y, but more favourable than generation BB.

Additionally, the research results indicated several gender differences. Women much more often than men declare a more favourable attitude towards information posted on social media and check SM opinions about places they want to visit. Positive opinions and comments in SM more often encourage women to go on tourist trips, and women more often resign from traveling due to negative comments posted on social media.

The obtained research results made it possible to formulate managerial and theoretical implications. The proposed recommendations concern the use of such marketing strategies in the field of the use of social media, in order to effectively influence the behaviour of individual generations, as regards the choice of their tourist destination, while maintaining sustainable development. The expectations and preferences of each generation group regarding the use of social media differ, so other marketing strategies are also needed to ensure the sustainable recovery of tourism. As the research of [80] shows, generations have different reasons for travelling; hence, other marketing activities are needed to encourage them to visit a given place. For example, for Generation Z, "the wisdom of the crowd" is important, so positive opinions and comments posted in SM are very important for them, and they will encourage representatives of this generation to go on holidays. In turn, Generations X and Y are more critical of information posted on social networks and do not show such enthusiasm as Generation Z. Basically, all generations admit that they check opinions and stories on the stay in tourist destinations in social media, but the BB generation takes a neutral stance in this regard. This is valuable information for marketers, so they can better reach their customers. As research by [51] shows, 45% of travel marketers emphasise that targeting travellers during a specific point along their path in purchasing is the greatest challenge for the coming years [51]. Therefore, information on the behaviour of potential tourists in social media at the stage of planning a holiday, collected in terms of age and gender, allows for more precise creation of promotional content. Since artificial intelligence (AI) allows advertising to be tailored to specific audiences, this information can provide opportunities to sell products and services in a travel destination. Consequently, software companies and researchers have developed several IT tools targeting specific customers, who need specific products and services, including an algorithm for the automatic generation of advertisements for Social Media platforms [81]. These tools can be used in the implementation of marketing activities aimed at the sustainable recovery of tourism. Data about users collected on social networks can be used to create profiles of potential customers. This will make it possible to reach them more effectively with a promotional message and encourage them to visit a given destination.

The interdisciplinary approach to the analysed problem encourages us to examine the possibilities of using practices from other industries and perspectives. Therefore, it is worth taking advantage of the recommendation technique [82] based on the identification of personality traits, moods, and emotions of a single user, starting from solid psychological observations recognized by the analysis of user behaviour within a social environment. This will be particularly important in the case of the necessity of exploring the emotional capacity of customers to travel during restart tourism. To increase the possibility of reaching potential tourists planning to visit places like cultural heritage it is worthwhile to use recommendation systems. According to [79], social data, can be used in several applications requiring a "social vision" of cultural items. Social media could be also used to obtain geographic data about the tourist places [83]. The recommendation system also enables exploiting the interactions among users and generated contents in one or more social media networks [84]. By obtaining such information, it will be possible to influence the behaviour of tourists and draw attention to the essence of sustainable development. According to [76] study, the word "sustainability" is presented in several posts: untouched beaches, lesser known tourist places, out-of-the way, and quiet corners free of mass tourism, and attractions and events where integration with and respect for the local community are ensured. Moreover, it will contribute to a positive impact on the environment, economy, and people. These activities can be performed through social marketing [85] and social media, e.g., user comments and opinions from common online social networks [79]. Thanks to them, it will be possible to engage tourists as stakeholders in the recovery of sustainable tourism and educate what behaviour they should display.

This article contributes to theory by extending the domain of sustainable development and tourism, as well as marketing theory. Moreover, it provides new knowledge about the preferences and behaviours of different generations at the stage of planning a tourist trip, as well as a new approach to the use of social media to restart tourism. Our study applied an original questionnaire.

The developed article has its limitations, which consist in the selection of the research sample. The number of responses from the BB generation was limited due to their limited use of new technologies. In the future, it is planned to conduct qualitative research that will allow examining the preferences of tourists and important factors determining the possibility of visiting a tourist attraction. Conducting research in focus groups is also considered. A weakness of the research is also its implementation in one country, which is why international research is planned to enable the identification of cultural factors, presented in [78,86], influencing the behaviour of representatives of particular generations.

Despite certain limitations, the presented article develops science by adding important elements concerning the generational differences in planning holidays. In addition, the article presents new ways of restoring tourism after the SARS-CoV-2 virus pandemic and various marketing activities.

**Author Contributions:** Conceptualization—B.H., A.K.; the review of literature—A.K., B.H.; the methodology—B.H., A.K., I.Z.; analysis of data—I.Z.; visualization—I.Z.; developed research results—I.Z.; conclusions and conduct the research—B.H., A.K., I.Z.; the final contents of the article and proofread—B.H. All authors have read and agreed to the published version of the manuscript.

**Funding:** This paper was published as part of statutory research at the Silesian University of Technology, Faculty of Organization and Management.

**Institutional Review Board Statement:** Not applicable.

**Informed Consent Statement:** Not applicable.

**Data Availability Statement:** Not applicable.

**Conflicts of Interest:** The authors declare no conflict of interest.

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
