# Peer review of "Social Media Usage by Different Generations as a Tool for Sustainable Tourism Marketing in Society 5.0 Idea"

_sustainability, doi:10.3390/su13031018_

Round 1

Reviewer 1 Report

Dear authors,

I read your article and it is interesting. The results show conclusions by types of generations.

However, some recommendations are needed to improve your article.

Thus, the "literature review" chapter is very extensive, which bores the reader and the essence of the content is lost. I recommend reducing the content and focusing on the title of each subchapter. Moreover, at point 2.4. The travel experience of different generations, I recommend dividing the text into subchapters. For each generation to be a subchapter.

Research hypotheses should be integrated into the text so that they are supported by the literature.

I have doubts about the sample size. Is it representative? The authors do not explain very clearly! Moreover, the sample is not balanced: BB = 2% and Z = 57%? Can the results be comparable?

At point 4.2.1. Generational and gender influences, the results must be presented in the order of the hypotheses in the text: H_2.1, H_2.2 ... H_2.8.

Some information from conclusions can be moved to results, because there are no conclusions.

The title of point 5 could be: Conclusions, limits of research and future research. The authors do not emphasize the novelty of their research. It is necessary in the introduction to focus on the novelty of the article and to resume in conclusions.

In conclusion, the article needs a much clearer structuring of information. As presented, it makes it difficult to read and understand by readers.

Best wishes!

Author Response

Manuscript Sustainability-1067561

Response to Reviewer 3

Dear Sir/ Madam,

We sincerely thank the Reviewer for constructive criticisms and valuable comments, which were of great help in revising the manuscript. We agree with all comments and suggestions. Accordingly, the revised manuscript has been systematically improved with new information and additional interpretations. We have highlighted the changes within the manuscript using the "Track Changes" function in Microsoft Word or writing comments in the margins. Our responses to the Reviewer’s comments are given below.

Comments from Reviewer

  • Comment 1:

The "literature review" chapter is very extensive, which bores the reader and the essence of the content is lost. I recommend reducing the content and focusing on the title of each subchapter.

Response:

We thank the Reviewer for pointing this out. We have revised and reduced some text at the “literature review”.

  • Comment 2:

Moreover,  at point 2.4. The travel experience of different generations, I recommend dividing the text into subchapters. For each generation to be a subchapter.

Response:

We are very grateful to the Reviewer for this comment. Text at point 2.4 was divided into subchapters.

  • Comment 3:

Research hypotheses should be integrated into the text so that they are supported by the literature.

Response:

We agree with the Reviewer that research hypotheses should be integrated into the text and should be supported by the literature. Our hypotheses are connected with differences between the generations in the scope of their use in planning a tourist trip. So we decided to put them after the part of literature in which the travel experiences of different generations were described.

  • Comment 4:

I have doubts about the sample size. Is it representative? The authors do not explain very clearly! Moreover, the sample is not balanced: BB = 2% and Z = 57%? Can the results be comparable?

Response:

We appreciate the Reviewer’s insightful suggestion and agree that the sample is not well balanced. When analyzing the sample structure, a small percentage of the BB generation participants can be seen. The number of responses from the BB generation was limited due to their limited use of new technologies. However, despite the small size of this subgroup of respondents, we decided to include it in the comparisons being at the center of our research problems. Therefore, we formulate all our conclusions bearing in mind the limitations resulting from the low presence of representatives of the BB generation. In the text of the article, in connection with the conclusions related to the confirmation or rejection of research hypotheses, we introduced additional comments referring to the limitations of our research related to the number of BB generations.

  • Comment 5:

At point 4.2.1. Generational and gender influences, the results must be presented in the order of the hypotheses in the text: H_2.1, H_2.2 ... H_2.8.

Response:

According to the suggestion, the results in section 4.2.1 have been presented in the order of the hypotheses in the text: H_2.1, H_2.2 ... H_2.8. Thus, the order of the rows in table 6 has also been changed.

  • Comment 6:

Some information from conclusions can be moved to results because there are no conclusions.

Response:

According to the Reviewers’ suggestions, we made corrections to the structure of the article. We split the discussion and conclusion sections. The discussion section was combined with the results.

  • Comment 7:

The title of point 5 could be: Conclusions, limits of research and future research. The authors do not emphasize the novelty of their research. It is necessary in the introduction to focus on the novelty of the article and to resume in conclusions.

Response:

We are very grateful to the Reviewer for this comment. We have corrected this part of the article and now there are five sections in our research manuscript: Introduction, Literature review, Materials and Methods, Results and discussion, Conclusions, limits of research and future research.

Moreover, we improved the discussion and conclusions section by connecting the findings with literature and extend the current knowledge from the new ones that were recommended by reviewers. We also described the novelties of our article in conclusions.

  • Comment 8:

In conclusion, the article needs a much clearer structuring of information. As presented, it makes it difficult to read and understand by readers.

Response:

We made corrections to the structure of the article in the conclusions section.

Moreover, we also corrected the article according to other reviewers:

  • we made corrections to the structure of the article,
  • we described the novelties of our article in conclusions,
  • we improved the article by adding the new literature which was recommended by reviewers and discussed it in the context of our research,
  • we corrected linguistic errors in the article
  • in tables 2-7 all comas have been replaced in dots,
  • we corrected all errors and small technical issues in the article

Once again, we thank you for your comments and hope our revisions now make the current version of the paper acceptable.

Authors

13th January 2021

Reviewer 2 Report

Dear Editor and Author,

Thank you for the opportunity to review the manuscript entitled: Social media usage by different generations as a tool for sustainable tourism marketing in Society 5.0 idea.

I like this paper very much, especially for the literature background, which I found an excellent part of the paper. Frankly speaking, only because of this background this paper is worth to be published. Thus my comments are just for further improving the clarity of the manuscript and better highlight the contribution of the author.

Overall, the topic of this paper is interesting and worthy of publication. The research part is clearly written, with good, concise prose, and also paper provides interesting findings.

There are two elements in this paper, where improvement will be appreciated. First is the discussion, and the second is technical issues. Also, in recent years social media like FB, IG, or YouTube become valuable tools both for information about tourism issues/routes/destinations, providing information of the current stage of the touristic project(s) and as a field of study in the context of tourism. Please consider adding below positions, the recent studies of article topic and make a discussion about it, in the context of your research (PS those are not mine, I just think that through them your paper will be more complete):

  1. Iglesias-Sánchez PP, Correia MB, Jambrino-Maldonado C, de las Heras-Pedrosa C (2020) Instagram as a co-creation space for tourist destination image-building: Algarve and costa del sol case studies. Sustain 12(7):1–26. https://doi.org/10.3390/su12072793
  2. Wijesinghe SNR, Mura P, Tavakoli R (2020) A postcolonial feminist analysis of official tourism representations of Sri Lanka on Instagram. Tour Manag Perspect 36(July):100756. https://doi.org/10.1016/j.tmp.2020.100756
  3. Latif K, Malik MY, Pitafi AH, Kanwal S, Latif Z (2020) If You Travel, I Travel: Testing a Model of When and How Travel-Related Content Exposure on Facebook Triggers the Intention to Visit a Tourist Destination. SAGE Open 10(2):215824402092551. https://doi.org/10.1177/2158244020925511
  4. Kozioł K, Maciuk K (2020) New heights of the highest peaks of Polish mountain ranges. Remote Sens 12(9):1446. https://doi.org/10.3390/rs12091446
  5. Paül Agustí D (2020) Mapping tourist hot spots in African cities based on Instagram images. Int J Tour Res 22(5):617–626. https://doi.org/10.1002/jtr.2360

D is exactly from the country you are in.

Please use information from your literature studies in your discussion. PS. Please split the discussion and conclusion. Usually, the reader will read an abstract and conclusion. That is why a conclusion is a very important part of the paper.

Technical issues. I think that most of them are made by mistake and can be quickly removed. I’m not sure but in some countries, perhaps in Poland also, the meaning of dot and comma is different than in the English language. Please take a look at Tables 2, 3, 4 and 5 (here also in the first column the text is not in English), 6, 7. You have a comma instead of a dot!

Also other small technical issues as line 167 or 232.

Author Response

Manuscript Sustainability-1067561

Response to Reviewer 1

Dear Sir/  Madam,

Thank you for allowing us to submit a revised draft of our manuscript titled “Social media usage by different generations as a tool for sustainable tourism marketing in Society 5.0 idea”. We are grateful to the reviewers for their insightful comments on our paper. We have been able to incorporate changes to reflect all of the suggestions provided by the reviewers. We have highlighted the changes within the manuscript using the "Track Changes" function in Microsoft Word or writing comments in the margins.

Here is a point-by-point response to the reviewers’ comments and concerns.

Comments from Reviewer

  • Comment 1:

Also, in recent years social media like FB, IG, or YouTube become valuable tools both for information about tourism issues/routes/destinations, providing information of the current stage of the touristic project(s) and as a field of study in the context of tourism. Please consider adding below positions, the recent studies of article topic and make a discussion about it, in the context of your research (PS those are not mine, I just think that through them your paper will be more complete):

  • Iglesias-Sánchez PP, Correia MB, Jambrino-Maldonado C, de las Heras-Pedrosa C (2020) Instagram as a co-creation space for tourist destination image-building: Algarve and costa del sol case studies. Sustain 12(7):1-26. https://doi.org/10.3390/su12072793
  • Wijesinghe SNR, Mura P, Tavakoli R (2020) A postcolonial feminist analysis of official tourism representations of Sri Lanka on Instagram. Tour Manag Perspect 36(July):100756. https://doi.org/10.1016/j.tmp.2020.100756
  • Latif K, Malik MY, Pitafi AH, Kanwal S, Latif Z (2020) If You Travel, I Travel: Testing a Model of When and How Travel-Related Content Exposure on Facebook Triggers the Intention to Visit a Tourist Destination. SAGE Open 10(2):215824402092551. https://doi.org/10.1177/2158244020925511
  • Kozioł K, Maciuk K (2020) New heights of the highest peaks of Polish mountain ranges. Remote Sens 12(9):1446. https://doi.org/10.3390/rs12091446
  • Paül Agustí D (2020) Mapping tourist hot spots in African cities based on Instagram images. Int J Tour Res 22(5):617-626. https://doi.org/10.1002/jtr.2360

Response:

We added the literature as suggested by the reviewer and discussed it in the context of our research. The discussion and conclusion section was improved.

  • Comment 2:

Please use information from your literature studies in your discussion.

Response:

We improved the discussion section by connecting the findings with literature and extend the current knowledge from the new ones that were recommended by reviewers.

  • Comment 3:

Split the discussion and conclusion. Usually, the reader will read an abstract and conclusion. That is why a conclusion is a very important part of the paper.

Response:

We made corrections to the structure of the article. We split the discussion and conclusion sections. According to instructions for the authors, section discussion was combined with results. There are five sections in our research manuscript: Introduction, Literature review, Materials and Methods, Results and discussion, Conclusions.

  • Comment 4:

Technical issues. I think that most of them are made by mistake and can be quickly removed. I’m not sure but in some countries, perhaps in Poland also, the meaning of dot and comma is different than in the English language. Please take a look at Tables 2, 3, 4, and 5 (here also in the first column the text is not in English), 6, 7. You have a comma instead of a dot!

Response:

In tables 2-7 all comas have been replaced in dots. In the manuscript, there are comments next to each table. In tables 4 and 5 text “Blogi i microblogi” (written in Polish) has been replaced by “Blogs and microblogs”.  In the manuscript, this change is saved in change tracking.

  • Comment 4:

Also other small technical issues as line 167 or 232.

Response:

We corrected all errors and small technical issues in the article.

Moreover, we also corrected the article according to other reviewers:

  • we described the novelties of our article in conclusions,
  • we improved the article by adding the new literature which was recommended by reviewers,
  • we corrected linguistic errors in the article.

Once again, we thank you for your comments and hope our revisions now make the current version of the paper acceptable.

Authors

8th January 2021

Reviewer 3 Report

The author investigate the use of social media by different generations as tools for tourist trips.

The proposed approach is interesting but there are some points that the authors have to better discuss.

The authors should be better described the novelties of their approach with respect to existing ones. In particular, the author should discuss limitation and cons that their approach aims to overcome at the end of the Related Works section. Furthermore, the authors should provide more details and discussion about the obtained results. The Discussion section also needs to be improved by analyzing the outcome of evaluation section.

I suggest to analyze also more recent approaches about the examined topics. In particular, I suggest to further investigate multimedia recommendation framework relying on social media data for attracting different generation in cultural heritage trip:

1) Kira: a system for knowledge-based access to multimedia art collections. In 2017 IEEE 11th international conference on semantic computing (ICSC) (pp. 338-343). IEEE.

2) Recommendation in social media networks. In 2017 IEEE Third International Conference on Multimedia Big Data (BigMM) (pp. 213-216). IEEE.

3) An emotional recommender system for music. IEEE Intelligent Systems.

Finally, I suggest to perform a linguistic revision.

Author Response

Manuscript Sustainability-1067561

Response to Reviewer 2

Dear Sir/ Madam,

Thank you for allowing us to submit a revised draft of our manuscript titled “Social media usage by different generations as a tool for sustainable tourism marketing in Society 5.0 idea”. We are grateful to the reviewers for their insightful comments on our paper. We have been able to incorporate changes to reflect all of the suggestions provided by the reviewers. We have highlighted the changes within the manuscript using the "Track Changes" function in Microsoft Word or writing comments in the margins.

Here is a point-by-point response to the reviewers’ comments and concerns.

Comments from Reviewer

  • Comment 1:

The authors should be better described the novelties of their approach with respect to existing ones. In particular, the author should discuss limitation and cons that their approach aims to overcome at the end of the Related Works section. Furthermore, the authors should provide more details and discussion about the obtained results. The Discussion section also needs to be improved by analyzing the outcome of evaluation section.

Response:

We described the novelties of our article at the end of the Related Works section and in conclusions.  The article contributes to theory by extending the domain of sustainable development and tourism, as well as marketing theory. Moreover, it provides new knowledge about the preferences and behaviours of different generations at the stage of planning a tourist trip, as well as a new approach to the use of social media to restart tourism.  Our study applied also an original questionnaire.

Our new approach will allow an in-depth analysis of travel planning behaviour by representatives of different generations in order to be able to better adapt promotional campaigns for them.

We improved the discussion section by connecting the findings with literature and extend the current knowledge from the new ones that were recommended by reviewers.

  • Comment 2:

 I suggest to analyze also more recent approaches about the examined topics. In particular, I suggest to further investigate multimedia recommendation framework relying on social media data for attracting different generation in cultural heritage trip:

  • 1) Kira: a system for knowledge-based access to multimedia art collections. In 2017 IEEE 11th international conference on semantic computing (ICSC) (pp. 338-343). IEEE.
  • 2) Recommendation in social media networks. In 2017 IEEE Third International Conference on Multimedia Big Data (BigMM) (pp. 213-216). IEEE.
  • 3) An emotional recommender system for music. IEEE Intelligent Systems.

Response:

We added the literature as suggested by the reviewer and discussed it in the context of our research.

  • Comment 3:

I suggest to perform a linguistic revision.

Response:

We corrected linguistic errors in the article.

Moreover, we also corrected the article according to other reviewers:

  • we made corrections to the structure of the article. We split the discussion and conclusion sections,
  • we added the literature as suggested by the reviewer and discussed it in the context of our research,
  • in tables 2-7 all comas have been replaced in dots,
  • we corrected all errors and small technical issues in the article

Once again, we thank you for your comments and hope our revisions now make the current version of the paper acceptable.

Authors

8th January 2021

Round 2

Reviewer 1 Report

Dear authors,

At point "4.2. Using SM at the planning stage of the trip", why did you make a subchapter "4.2.1. Generational and gender influences"? In this context, you need a subchapter 4.2.2 .. Please clarify this point.

After solving this requirement, I consider that the article meets the conditions to be published.

Good luck!

Author Response

Manuscript Sustainability-1067561

Response to Reviewer 3

Dear Sir/ Madam,

We thank the Reviewer for their time spent carefully reviewing the manuscript, and in their opinions regarding the science and presentation of the material. We have been able to incorporate changes to reflect all of the suggestions provided by the Reviewer. We have highlighted the changes within the manuscript using the "Track Changes" function in Microsoft Word or writing comments in the margins. Our responses to the Reviewer’s comments are given below.

 Comments from Reviewer

  • Comment 1:

At point "4.2. Using SM at the planning stage of the trip", why did you make a subchapter "4.2.1. Generational and gender influences"? In this context, you need a subchapter 4.2.2 .. Please clarify this point.

 Response:

As suggested by the Reviewer, we have removed subchapter 4.2.1. We have moved the entire text from this section to chapter 4.2. Thus, chapter 4.2 has no sub-chapters.

Once again, we thank you for your comments and hope our revisions now make the current version of the paper acceptable.

Authors

15th January 2021

Reviewer 3 Report

I think that the authors have addressed all my concerns. 

Author Response

We would like to thank the Reviewer for accepting our revisions.